# Molecular Mechanisms Associated with Aging Kidneys and Future Perspectives

**DOI:** 10.3390/ijms242316912

**Published:** 2023-11-29

**Authors:** Min-Jee Jo, Joo-Kyung Lee, Ji-Eun Kim, Gang-Jee Ko

**Affiliations:** 1Department of Internal Medicine, Korea University College of Medicine, Korea University Guro Hospital, Seoul 08308, Republic of Korea; pinkle@korea.ac.kr (M.-J.J.); jjoo.leah@gmail.com (J.-K.L.); beeswaxag@naver.com (J.-E.K.); 2Institute of Convergence New Drug Development, Korea University College of Medicine, Seoul 08308, Republic of Korea

**Keywords:** aging kidney, glomerulosclerosis, inflammation, cell cycle, oxidative stress

## Abstract

The rapid growth of the elderly population is making the need for extensive and advanced information about age-related organ dysfunction a crucial research area. The kidney is one of the organs most affected by aging. Aged kidneys undergo functional decline, characterized by a reduction in kidney size, decreased glomerular filtration rate, alterations in renal blood flow, and increased inflammation and fibrosis. This review offers a foundation for understanding the functional and molecular mechanisms of aging kidneys and for selecting identifying appropriate targets for future treatments of age-related kidney issues.

## 1. Introduction

Aging causes gradual organ changes over time that eventually lead to senescence-related disease and death [1]. In recent years, the elderly population has increased exponentially, and the number of people between the ages of 66 and 84 is expected to exceed 61 million by 2030. Therefore, research on the aging process itself and related health issues, age-related disease progression, and disability among elderly people is critically important [2,3].

It is widely recognized that the kidney, which is responsible for blood purification and blood pressure management, as well as governing homeostatic processes such as salt and water, acid and base, and bone and mineral balances [4,5], undergoes structural and functional changes with age. Kidney mass increases from about 50 g at birth to more than 400 g during adolescence; however, it decreases to less than 300 g by age 90 [6]. Moreover, nephrons, the structural and functional units of the kidney, are lost with aging. It is estimated that people aged 70 to 75 have 48% fewer nephrons than people aged 18 to 29. In a study of kidney donors, individuals aged 18 to 29 years had an average of 990,661 non-sclerotic glomeruli per kidney, decreasing to 520,410 per kidney in donors aged 70 to 75 years. On the other hand, an average of 16,614 sclerotic glomeruli per kidney was observed in donors aged 18 to 29 years, and that number increased to 141,714 per kidney in donors aged 70 to 75 years [7]. Renal plasma flow remains constant at 600 mL/min until about age 40, after which it decreases by approximately 10% every decade [8]. These age-related alterations enhance the risk of both acute and chronic kidney disease for elderly people [9].

Understanding the decline in renal function caused by aging is an emerging area of research aimed at discovering potential treatments for kidney diseases. This review focuses on aging kidneys and their underlying mechanisms and features, including an analysis of molecular mechanisms reported in clinical data and animal models.

## 2. Functional Changes in Aging Kidneys and Measuring Kidney Function in Elderly People

### Aging and Decline of Kidney Functions

The prevalence of chronic kidney disease (CKD) has been increasing for decades, particularly among elderly people. According to an analysis by the National Health and Nutrition Examination Survey 2001–2020, the prevalence of CKD between 2017 and 2020 was 20.1% and 42.6% among people aged 60–69 years and older than 70 years, respectively, which was much higher than the 14.8% reported for the total population [10]. Although prevalence varies between regions and studies, a meta-analysis that included one hundred manuscripts also reported that the global prevalence of advanced CKD, defined as CKD 3–5, increased with age: 27.9% among 70-year-olds but only 8.1% in the total population [11]. More than 50% of people older than 75 years are thought to have reduced kidney function. Considering that many countries are experiencing an exponential increase in the elderly population, CKD prevalence is expected to increase rapidly, and its management is emerging as an important issue. Already, the proportion of patients starting dialysis who are aged 65 or older exceeds 50% [12], and the cumulative costs for dialysis treatments are a significant burden on the healthcare system. CKD is also associated with an increased risk of cardiovascular events and hospitalization [13]. Therefore, the increasing prevalence of CKD, especially among elderly people, will greatly affect healthcare budgets and should be carefully addressed. We summarize epidemiological studies of CKD prevalence in large-scale cohorts in Table 1 [14,15,16,17,18,19,20,21].

Kidney function declines with age for several structural and clinical reasons. Structural changes, such as a decrease in the number of functional units and reduced kidney blood flow, limit the capacity of the kidneys to eliminate waste products efficiently, which manifests as a constant decrease in filtration rate after 40 years of age [8,22,23]. Intraglomerular hypertension caused by the loss of glomeruli results in the hypertrophy and hyperfiltration of the remaining nephrons [24], which can begin a vicious cycle of insults that culminates in further nephron loss and irreversible kidney damage. The high prevalence of underlying diseases in CKD, such as hypertension and diabetes, is another major factor contributing to the problem. Moreover, the elderly population has a particularly high chance of exposure to the various risks that can cause acute kidney injury (AKI), with elderly people often experiencing delayed or incomplete recovery after AKI [25]. Dehydration is commonly observed in elderly individuals due to reduced thirst sensation and appetite, and chronic dehydration can place additional stress on the kidneys. Polypharmacy happens easily in elderly people, and certain medications, especially those for neuro-muscular-skeletal diseases or infectious diseases, can damage kidney function directly or indirectly. The contrast medium used to diagnose and treat diseases common in elderly people, such as cancer and cardio-cerebral vascular diseases, is also nephrotoxic. The risk to the kidneys can be escalated by more frequent exposure and a reduced capacity for handling toxic metabolites with age.

## 3. Molecular Mechanisms in Aging Kidneys

Complex changes associated with multiple molecular alterations in various parts of the kidney are implicated in the decline of kidney dysfunction with age. Figure 1 presents a schematic summary of changes in the molecular mechanisms of aged kidneys. Each change can occur as an interaction among different kidney parts. The detailed mechanisms of those molecular alterations are described below.

### 3.1. Glomerular Changes in Aging Kidneys

The glomerulus is responsible for filtering the blood, an essential role of the kidney. It is a core element of the nephron, the fundamental structural and functional unit of the kidney. Mesangial cells and the mesangial matrix are located between capillaries, supporting the endothelial, epithelial, and three-layered basement membranes that constitute the capillary wall [26]. With age, the glomerular basement membrane (GBM) widens, and the mesangial compartment increases, leading to the expansion of the glomerulus, which is associated with glomerular hyperfiltration. Thickening of the GBM, along with an accumulation of extracellular matrix proteins, eventually leads to progressive glomerulosclerosis [27]. These alterations in the aging glomerulus reduce the glomerular filtration rate and the ability to concentrate urine, potentially causing glomerular disease [28]. Glomeruli from 27-month-old C57B6 mice showed an increased tuft area and a decreased number of P57-positive podocytes and PAX2-positive glomerular parietal epithelial cells (PECs). This resulted in a significantly lower cell density in the glomeruli compared to that found in the glomeruli of 3-month-old mice, with the decrease being more prominent in the paraspinal glomeruli than in the outer cortical glomeruli [27]. Those changes were associated with an increase in cells that were positive for CD44, an activation marker of PEC cells, and an increased expression of phosphorylated ERK and extracellular matrix proteins, such as heparan sulfate proteoglycan. Those changes, in turn, resulted in the transformation of PECs through an increase in epithelial to mesenchymal transition markers, such as collage type IV, vimentin, and α-smooth muscle actin. Activated PECs such as pericytes also manifested an increase in platelet-derived growth factor receptor-β, neuronal/glial antigen 2, CD146, and Notch 3, a profibrotic marker. Thus, aging causes substantial changes in PECs, including the loss of their role as progenitors of podocytes, as the expression of laminin decreases, until they begin to exhibit characteristics of transition cells, resembling pericytes and mesenchymal fibroblasts, particularly in the juxtamedullary glomeruli. This suggests that PECs could play a key role in age-associated glomerulosclerosis and interstitial fibrosis [27].

Podocytes play a significant role in glomerular function by maintaining and supporting the GBM. In aged podocytes in mice, glycogen synthase kinase 3 (GSK3β) was found to be overexpressed and overactivated [29]. GSK3 is a serine/threonine protein kinase with ubiquitous and constitutively active expression. It plays a pivotal role in various signaling pathways associated with tissue injury, repair, and cellular regeneration [30]. The role of GSK3β in podocyte aging was demonstrated using podocyte-specific GSK3β knockout (KO) mice. Primary cultured podocytes from GSK3β KO mice showed mitigated cellular senescence and expression of the senescence-associated secretory phenotype (SASP). Those authors argued that GSK3β modulated senescence signals by controlling the phosphorylation of p16^INK4a^ and p53, known as senescent activator genes, in podocytes, which led to extracellular matrix accumulation in the glomeruli and renal interstitium, increased numbers of sclerotic glomeruli, broadening of the podocyte foot, and amplified expression of profibrotic factors such as fibronectin [29]. Another study revealed the role of sirtuin 1 (SIRT1) expression in the podocytes of aging kidneys. Podocyte-specific SIRT1 KO in mice aggravated aging-associated glomerulosclerosis and albuminuria and was associated with an increase in cellular oxidative stress, measured by urinary 8-hydroxy-2′-deoxyguanosine (8-OHdG), and reduced podocyte maturation. SIRT1-KO mice showed reduced expression of peroxisome proliferator activated receptor γ (PPARγ) and PPAR-α coactivator-1 (PGC1α) in the glomerulus, which accelerated the acetylation of PGC1α and forkhead box O (FOXO)_3_, FOXO_4_, and p65 NF-κB [31]. The role of programmed cell death protein-1 (PD-1) in podocyte aging was also revealed. PD-1 overexpression occurred in the podocytes of aged mice and correlated with a decrease in eGFR and an aggravation of glomerulosclerosis and the vascular arterial intima-to-lumen ratio. Blocking PD-1 signaling with a neutralizing anti-PD-1 antibody reduced podocyte senescence and increased podocyte lifespan, which was manifested as increased cell density and accompanied by a marked decrease in multiple inflammatory cascades, such as Toll-like receptors, IFN-α and -γ, interleukin (IL)-6/JAK/STAT, IL-2/STAT, and KRAS [32]. In another experimental study, an RNA-sequencing analysis of podocytes from aged mice showed inflammatory phenotype changes, particularly increases in the nod-like receptor protein 3 (NLRP3) inflammasome (NLRP3, caspase-1, IL-1β, and IL-18), IL-2/STAT5 signaling, IL-6, TNF, and IFN-γ. Experimentally induced focal glomerulosclerosis was augmented in aged mice [33]. A clinical study supported these finding, as an increased expression of NLRP3 inflammasomes was observed in human kidney samples obtained from an 82-year-old patient compared with samples from a 23-year-old patient. The pharmacological inhibition of NLRP3 by MCC950 or gene deletion in NLRP3-null mice improved podocyte lifetime and the glomerular ultrastructure. A higher expression of NLRP3 in human glomeruli was associated with lower podocyte density, increased glomerular volume, and total glomerulosclerosis [33]. The knockout of CAAT enhancer-binding protein α (C/EBPα) in podocytes also aggravated senescence through the AMPK/mTOR pathway, which could lead to glomerulosclerosis and albuminuria [34]. Experiments targeting these markers should be conducted to explore treatments for podocyte senescence.

### 3.2. Tubular Changes in Aging Kidneys

Aging is associated with tubular atrophy, malfunction, and decreased sodium reabsorption and potassium excretion. Various molecular mechanisms are thought to be involved in those changes. Figure 2 summarizes the molecular mechanisms associated with aged tubules.

#### 3.2.1. Cell Cycle

Cellular senescence is characterized by cell cycle arrest in the G1 or G2 phase, which restricts the proliferation of cells damaged by both endogenous and exogenous stresses. However, this process can hinder the healing process, which is primarily observed in the tubules of kidneys [35,36]. Under normal conditions, only one percent of proximal tubular cells express proliferation markers. However, in response to an acute insult, the renal tubular epithelium initiates a burst of proliferation for repopulation and healing. Age-related alterations in the expression of regulatory proteins for the cell cycle have been found in aging kidneys, and the proliferative potential of tubular cells declines with age. The mRNA and protein expressions of p21, known as a cell cycle inhibitor, were significantly higher in 24-month-old mice than in 2-month-old mice, and the p21 concentration also increased with age in plasma [37]. Cdkn1a transcript-specific variant 2 is a sensitive marker for cellular senescence and associated with cell cycle regulation. An enhanced mRNA expression of *Cdkn1a variant 2* was observed in aged kidneys [38]. The cell cycle protein cyclin D1 was higher in the kidneys of old mice than young mice, and the correlation between age and tubular epithelial cyclin D1 was also observed through transplant biopsies and tissue sections of nephrectomy in human kidney samples from old (76 years old) and young (36 years old) patients [39]. Because cyclin D1 expression is already high in old kidneys, additional mitogenic stimuli might not increase the expression properly, potentially causing a disruption in cell cycle progression from the late G1 phase into the S phase with age. The expression of cell cycle regulators that inhibit the cell cycle, such as p16^INK4a^, COX-1, and COX-2, was also elevated in aging human kidneys [40]. The importance of p16^INK4a^ expression in the senescence of somatic cells has been demonstrated in several studies. The expression of p16^INK4a^ was found to be high in the kidneys of old rats [41], and it is associated with age-related renal fibrosis through its interactions with Wnt/β-catenin signaling, Klotho, and renin-angiotensin system activation [42]. In an accelerated aging model induced by D-galactose (d-gal) treatment, p16^INK4a^ expression increased, and was associated with the expression of γH2AX and p19ARF. This association caused mitochondrial dysfunction and the downregulation of PGC-1α, transcription factor A mitochondria, and oxidative phosphorylation proteins [42]. In another experimental study, B lymphoma Mo-MLV insertion region 1 homolog (Bmi-1) deficiency resulted in premature renal-aging phenotypes with increased apoptosis and SASP, tubulointerstitial fibrosis, and tubular atrophy. The deletion of p16^INK4a^ mitigated renal damage from the premature aging induced by Bmi-1 deficiency [43].

#### 3.2.2. Pro-Fibrotic Pathway

The progression of kidney fibrosis is the most prominent feature of aging kidneys. Senescent cells release a variety of pro-fibrotic molecules, most notably SASP components, including IL-1β, IL-6, IL-8, C-X-C motif chemokine ligand 1 (CXCL1), and transforming growth factor-β1 (TGF-β1) [44]. Another well-known senescence suppressor gene, Klotho, is mainly expressed in the cell surface membranes of proximal and distal renal tubules [45], and it can modulate kidney fibrosis. In 24-month-old mice, Klotho expression was low. However, Klotho overexpression in transgenic mice with the ubiquitous human elongation factor-1-a promoter (EFmKL46) protected the kidneys from age-associated fibrosis, probably by modulating TGF-β1, SMAD family member 4 (Smad4), connective tissue growth factor, and the phosphorylation of Smad2/3 [46]. Several proteins associated with the Wnt/β-catenin signaling pathway, including β-catenin Wnt1, Wnt10b, β-catenin, and active β-catenin, were more highly expressed in the kidneys of 24-month-old mice than 7-month-old mice. Additionally, providing secretory Klotho to old mice via plasmid pV5-sKlotho significantly inhibited the expression of Wnt10b and active β-catenin. These changes attenuated kidney fibrosis in aged mice by decreasing the expression of fibronectin and α-SMA [42]. Another experimental study of aged kidneys revealed increased hypoxia-inducible factor 1 subunit α (HIF-1α) expression, with the acetylation of which being associated with decreased SIRT1 expression. The protein expression and activity regulated by HIF-1α, such as BCl-2 adenovirus E1B-interacting protein 3, carbonic anhydrase 9, Snai1, and TGF-β1, was also high in aged mice, suggesting that the HIF-1α deacetylation induced by SIRT1 might be protective against tubulointerstitial damage in aged kidneys. To reveal the specific role of HIF-1α in the kidney tubules of aged mice, paired box 8–reverse tetracycline-responsive transactivator-based conditional HIF-1α transgenic mice with kidney tubule-specific HIF-1α overexpression were examined. Those mice showed accelerated kidney damage, including a high level of collagen deposition and an increased number of apoptotic cells in the tubulointerstitium, which supports the role of SIRT1 and HIF-1α in tubular cell aging [47].

#### 3.2.3. Oxidative Stress

An imbalance in the generation of reactive oxygen species (ROS) leads to oxidative stress, and it is widely accepted that the accumulation of oxidative stress plays an important role in age-related functional decline [48]. It has been demonstrated that the activation of nuclear erythroid-related factor 2 (NRF2), which is crucial for regulating the redox system, can improve aging-related phenotypes. The role of NRF2 in kidney aging was examined with a combination of NRF2 KO (NRF2−/−) and Keap1 KO (Keap1 KD mice) mice crossed with a premature aging model, a-Klotho-deficient mice (Kl−/−). In the Keap1 KD:: Kl−/− mice exhibiting persistent systemic NRF2 activation, the gene expression of antioxidant enzymes, including *Nqo1*, *Txnrd1*, *Gsta4*, and *Gstp1*, was elevated, and oxidative stress, examined via 8-OHdG staining, was decreased, which improved the renal aging phenotype by significantly attenuating tubular fibrosis and calcification [49]. The decreased expression of NRF2 in older mice was demonstrated in other studies [50,51], and the induction of NRF2 expression with sulforaphane treatment reduced oxidative stress and vascular calcification in aging mice [51]. Higher oxidative stress levels, measured by malondialdehyde, and lower levels of total antioxidant capacity, such as superoxide dismutase (SOD) and catalase, were shown in aged rats, and those effects were suggested to be linked to reduced SIRT1 and NRF2 expression [52]. Another experimental study demonstrated that old mice showed lower SIRT1 and Klotho expression than young mice. In that study, concanavalin A (Con A) was used to produce renal oxidative stress and progressive glomerulosclerosis in aging mice. Following Con A stimulation, SOD activity, glutathione, Klotho, and SIRT1 levels were all reduced, which resulted in 8-hydrogen oxidation. The levels of 8-OHdG and ROS were found to be significantly elevated. Resveratrol treatment has been demonstrated to reduce renal oxidative stress and prevent Con A-induced progressive glomerulosclerosis through SIRT1-mediated Klotho expression in old mice [53]. The overexpression of glutathione peroxidase-1, one of the most important antioxidant enzymes, in aged kidneys was found to independently reduce glomerulosclerosis and interstitial fibrosis in old mice through antioxidant activity with NRF2 and Klotho signaling [54].

Excess oxidative stress contributes to endoplasmic reticulum (ER) stress in aged mice. A significant increase in oxidized proteins and advanced glycation end products was noted, along with a demonstrated reduction in the ratio of oxidized glutathione (GSH/GSSG) in the kidneys of old mice [55]. An accumulation of unmodified proteins in the ER induces a stress response called the unfolded protein response (UPR). The loss of X-box binding protein 1 splicing and reduced pancreatic ER kinase phosphorylation, both known to regulate the UPR, resulted in abnormal ER stress in aged mice compared with young mice [56].

In a study comparing 6-, 12-, 18-, and 24-month-old mice, PPARα and the fatty acid oxidation pathway in renal tubules declined with aging. The expression of miR-21, an miRNA known to inhibit PPARα translation, was substantially upregulated, and the ablation of PPARα triggered an accumulation of renal lipids and the deterioration of renal fibrosis [57].

#### 3.2.4. Inflammation

Immune system dysfunction and alterations in the inflammatory pathway are crucial components of aging processes that promote tissue degradation [58]. Immunosenescence is characterized by impaired neutrophilic phagocytic capacity, the depletion of B cells, naïve T cells, and dendritic cells, as well as the accumulation of aberrant phenotypes. Immune dysfunction associated with aging leads to chronic low-grade inflammation through the impairment of proper immune protection [59,60].

Increased secretion of pro-inflammatory cytokines, such as TNF-α, IFN-γ, and IL-12, and increased mRNA expression of pro-inflammatory genes, including *MCP-1*, *RANTES*, *MIP-2*, *CXCL1*, and *ICAM-1*, were found in the kidneys of aged mice [61,62]. The levels of IL-1 and MCP-1 were upregulated in the kidneys of old mice, accompanied by an increase in arginase-II (Arg-II), which is exclusively expressed in renal proximal tubules. Additionally, TNF-α, IL-1β, IL-6, MCP-1, VCAM-1, ICAM-1, macrophage marker f4/80, and inducible nitric oxide synthase were expressed at higher levels in aged mice compared to young mice. Age-related IP-1, MCP-1, VCAM-1, and macrophage accumulation in mouse kidneys was decreased by Arg-II deficiency [63]. An analysis based on RNA sequencing of the kidneys revealed a higher expression of Arg-II, a type II L-arginine: ureohydrolase, along with upregulation of NF-κBIZ, IL-6, and MCP-1. Cytokines/chemokines involved in tubulointerstitial fibrosis, including IL-1β, VCAM-1, and TGF-β1, were also elevated. Additionally, macrophage infiltration was increased in 24-month-old rats compared to 6-month-old rats [63,64]. Among the cell types involved in inflammation, two cell populations, a mesenchymal stromal cell subunit (kSMC)-expressing CD73 and a monocyte-derived Ly6C_CCR2_ macrophage subset, were upregulated with aging [65]. Aged CD73^+^ kMSCs exhibited senescence-related features related to low proliferative rates, and elevated DNA damage foci and CCL2 expression. Senescent CD73^+^ kMSCs were found to trigger the release of inflammatory cytokines, unlike young CD73^+^ kMSCs [65]. When transcriptomes and epigenomes of the kidneys were profiled in young and aged mice, inflammation-associated genes, including *Cd74*, *Cxcl13*, and extended *Fc receptor-like 5*, were in the set of genes that were upregulated, and *phosphoenolpyruvate carboxykinase 1*, *3-hydroxy-3-methylglutaryl-CoA reductase*, and *Cyp2c50* were in the set of genes that were downregulated in aged mice [66]. The recruitment of immune cells and pro-inflammatory gene expression were higher in the kidneys of old mice than young mice [66].

### 3.3. Vascular Changes with Age

Blood vessels undergo various changes with age. Renal blood flow declines and renal vascular resistance increases with advancing age as a result of arteriolar vasoconstriction. These changes ultimately impact GFR [67]. In addition, changes in vascular tone occur in response to vasoconstrictors and vasodilators [68].

Blood flow within the glomeruli also changes with aging. The effective blood flow in the kidneys progressively decreases from the glomeruli of the outer cortex due to increased post-glomerular vascular resistance. Shunts between the afferent and efferent arterioles allow for direct flow, bypassing the glomeruli around the juxtamedullary zone [27]. In an experimental study with aged rats, afferent arteriole resistance was reduced. This led to an independent increase in the glomerular capillary hydraulic pressure with changes in systemic pressure, and contributed to glomerulosclerosis [69].

The renin–angiotensin–aldosterone system (RAAS) governs extracellular fluid and blood pressure, and plays an essential role during kidney aging. An examination of early-phase (3-month-old) and late-phase (12-month-old) aging in rats revealed a decline in plasma renin concentration and a reduction in *renin* mRNA expression with aging [70]. Old mice (24 months old) had lower levels of *renin* mRNA and fewer cells in the renin lineage than young mice (3.5 months old) [71]. The intrarenal vascular tone in an aging kidney can be modulated by the reduction in renal angiotensin-converting enzyme (ACE) levels associated with aging [72]. In old rats, tubulointerstitial fibrosis and an increase in the expression of fibronectin and TGF-β were accompanied by an increase in the angiotensin II type 1 receptor (AT1R)/AT2R ratio, renin receptor expression, and cortical ACEs [73]. The pressure response to angiotensin II (Ang II) was delayed in young mice, whereas it immediately increased systolic blood pressure in old mice. Renal ATR expression also increased with age. The expression of the AT1R/AT2R ratio in mesenteric arteries increased with age, which could contribute to the enhanced responses to Ang II seen in old mice. Moreover, old mice produced more superoxide in the mesenteric artery and thoracic aorta in a nicotinamide adenine dinucleotide phosphate oxidase 2 (NOX 2)-dependent manner [74].

The administration of resveratrol to control RAAS in aged kidneys reduced the Ang II/AT1R axis and enhanced the AT2R/Ang 1–7/Mas receptor (MasR) axis. Moreover, resveratrol treatment decreased the expression of NOX 4, 8-OHdG, collagen IV, and fibronectin while increasing the expression of endothelial nitric oxide synthase and SOD 2. These changes inhibited the activation of Ang II and MasR, mitigated renal damage, and lessened albuminuria [75]. The question of whether RAS increases or decreases as kidneys age is still debated. Nevertheless, RAS undeniably plays a significant role in the aging of kidneys, contributing to an imbalance in the renin–angiotensin–aldosterone system (RASS) [76].

## 4. Metabolic Changes in Aging Kidney

Although research on the causes, renal metabolites, and metabolic pathways that affect renal metabolic dysfunction due to aging has been insufficient, it is known that renal aging induces a variety of alterations in biological metabolic processes [77]. A high-resolution metabolomics analysis examined age-related alterations in kidneys and urine in 3-month-old and 24-month-old mice. A clear distinction appeared in the metabolic profiles of the two groups [78]. Different activations of the metabolic pathway related to D-glutamine and differences in D-glutamate metabolism, purine metabolism, and the TCA cycle were identified between the kidneys of young and old mice. The TCA cycle (also known as the Krebs or citric acid cycle), histidine metabolism, and pyruvate metabolism in urine also differed between the young and old mice. Furthermore, the FOXO, glucagon, and HIF-1α signaling pathways were associated with renal and urine metabolism [78]. The activities of various citric acid cycle enzymes in mitochondrial preparations from young (6-month-old), middle-aged (16-month-old), and elderly (24-month-old) mouse kidneys were altered during the aging process. Among the citric acid cycle enzymes, aconitase revealed the most notable reduction in activity related to age. The activity of ketoglutarate dehydrogenase decreased to a certain extent, and the activity of NADP+-isocitrate dehydrogenase increased. Additionally, as the mice aged, the NADPH/NADP+ molar ratio decreased. Citrate enzyme modifications imply that a bioenergetic deficit occurs during the aging process [79].

Mitochondria are complex organelles for fuel metabolism that produce energy in the form of adenosine triphosphate (ATP) [80]. The kidney is a mitochondria-rich organ with a high metabolic rate, and mitochondrial biogenesis is considered to be a significant mediator in aging kidneys [80,81]. A comparison of the kidneys of young, middle-aged, and old mice showed a functional decrease in mitochondrial electron transport chain complexes I, II, IV, and V in old mice [82]. The expression of mitochondrial proteins, including phosphorylated PGC-1α (p-PGC-1α), outer mitochondrial membrane protein translocase 20, cytochrome b, and COX1, is mostly lost as mtDNA declines with age [42]. The deletion of mitoregulin (Mtln), located in mitochondria and contributing to oxidative phosphorylation and fatty acid metabolism, might accelerate aging. Old Mtln KO mice exhibited an increased frequency of renal proximal tubule degeneration and a reduced glomerular filtration rate compared with wild-type mice [83]. Phosphatase and tensin homolog-induced kinase 1 (PINK1) is known to regulate mitochondrial function. PINK1 deficiency enhances kidney fibrosis, promotes cellular senescence of renal tubular epithelial cells, and increases SASP. The dysregulation of mitochondria in PINK1 deficiency was most apparent in 24-month-old mice. A gene expression analysis using RNA sequencing revealed increased inflammatory responses and activation of the cyclic GMP-AMP synthase stimulator of interferon genes pathway in old mice [84]. Cannabinoid receptor 2 (CB2) is also known to play an important role in the mitochondrial dysfunction of kidney tubules in aging kidneys. CB2 levels were upregulated in the kidneys of 24-month-old mice. An accelerated aging mouse model through d-gal treatment produced a decrease in mitochondrial mass. The genetic deletion of CB2 in the d-gal-induced mice suppressed β-catenin signaling activation and restored mitochondrial integrity and ATP production. The decreases in tubular cell density and kidney fibrosis seen in the accelerated aging models were inhibited in CB2 KO mice [85].

## 5. Future Perspectives

The kidney is one of the organs most affected by aging. Despite its importance and the absence of a clearly presented new concrete treatment strategy for aging kidneys, a thorough understanding of the molecular pathways active in aging kidneys might help elucidate beneficial strategies. Although a significant amount of research on kidney aging has been conducted, much remains to be done. To fully understand and confront aging, novel aging molecules and research methodologies should be discovered. For example, drugs newly developed to treat other diseases, such as cancer, have gained attention for their potential to prevent renal damage. PD-1/PD-L1 inhibitors, known as immune checkpoint inhibitors, have shown therapeutic promise for the treatment of various cancers and might also offer renal protection, as we reviewed above. PD-1 signaling is being explored for its therapeutic potential in clinical trials focused on renal aging [32,86]. Moreover, it is crucial to screen and uncover aging factors using recent technological advances, such as bulk and single-cell RNA sequencing technologies and the recently introduced spatial transcriptomic technology, which isolates RNA from a particular region in situ. Those approaches can avoid the drawbacks of direct viewing and make comprehensive analyses accessible. In particular, single-cell RNA sequencing with a spatial consideration could elucidate target molecules in specific parts of aged kidneys. For example, a single-cell RNA sequencing analysis revealed tubular cells changing into an inflammatory phenotype and associated target molecules in aged murine and human kidney tissues [87]. A single-cell RNA sequencing analysis of aged mouse kidneys revealed seven subtypes of renal endothelial cells (ECs), with glomerular ECs and angiogenic ECs being the most aged. Furthermore, a pro-coagulant and pro-inflammatory microenvironment manifested in aged glomerular ECs [88].

To alleviate the deterioration of renal function in aging kidneys, it is necessary to detect the decline in renal function at an early stage, and factors that affect those early changes must be found. To date, creatinine has been used as the standard test to monitor kidney dysfunction, but because muscle mass is low in elderly people, the ability of creatinine to reflect early decline in renal function is limited. Therefore, efforts have been made to discover a more sensitive biomarker. The most fully verified substance to date is cystatin C, but confirmative results have yet to be published. Efforts to detect kidney damage in the early stages using substances related to aging should continue.

Metabolites are the various byproducts of metabolic processes. Metabolites, genomes, and proteomes are organically linked in vivo. Determining molecular pathways requires an understanding of those relationships. Computational metabolomic analyses such as pharmaceutical-based metabolic modeling or in silico metabolic analysis could play a crucial role in the study of aging, but they have not played a significant role yet. Ultimately, multi-omics research from a variety of perspectives on the genome, transcriptome, and metabolome might help to identify diagnostic and therapeutic markers for aging kidneys and should be actively considered in the future.

## 6. Conclusions

Complex changes in various parts of the kidney are implicated in kidney dysfunction in elderly people. Changes that occur sequentially with aging are connected organically to molecular alterations in adjacent tissues. An individual’s sensitivity and vulnerability to those molecular alterations might be linked to the degree of decline in kidney function with age. A schematic description of changes that happen in the various tissues of aged kidneys is presented in Figure 3. Because aging kidneys are clinically associated with increased vulnerability to various renal diseases, studies to prevent and treat kidney dysfunction in aging kidneys should continue. Research on the molecular underpinnings of aging in kidneys is also incomplete, and distinctive biomarkers for aging kidneys are needed. The most recent research provides an overview of the functional and molecular mechanisms of aging kidneys and might ultimately be used to develop kidney disease diagnostic, therapeutic, and prognostic prediction technologies.

## Figures and Tables

**Figure 1 ijms-24-16912-f001:**
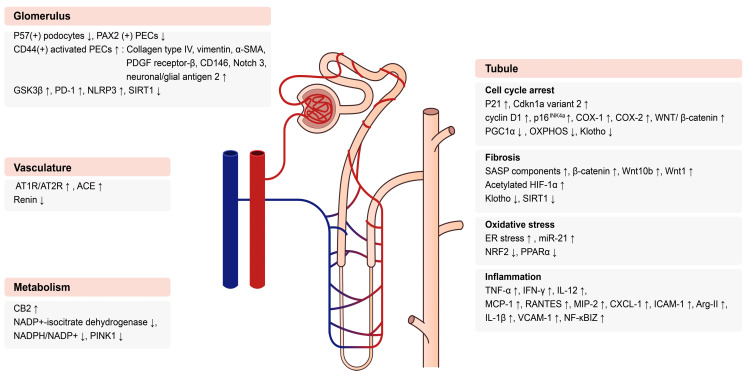
Changes in molecular mechanisms of aged kidneys. Abbreviations: PAX2, paired box gene 2; PEC, parietal epithelial cell; SMA, α-smooth muscle actin; PDGF, platelet-derived growth factor; GSK3, glycogen synthase kinase 3; PD-1, programmed cell death protein-1; NLRP3, nod-like receptor protein 3; SIRT1, sirtuin 1; AT1R, angiotensin II type 1 receptor; AT2R, angiotensin II type 2 receptor; ACE, angiotensin-converting enzyme; CB2, cannabinoid receptor 2; NADP, nicotinamide adenine dinucleotide phosphate; PINK1, phosphatase and tensin homolog-induced kinase 1; COX-1, cyclooxygenase 1; COX-2, cyclooxygenase2; PGC1α, PPAR-α coactivador-1; OXPHOS, oxidative phosphorylation; SASP, senescence-associated secretory phenotype; HIF-1α, hypoxia-inducible factor 1 subunit α; ER, endoplasmic reticulum; NRF2, nuclear erythroid-related factor 2; PPARα, peroxisome proliferator activated receptor α, TNF- α, tumor necrosis factor-α; IFN-γ, interferon γ; MCP-1, monocyte chemoattractant protein 1; RANTES, regulated upon activation, normal T cell expressed and presumably secreted; MIP-2, macrophage inflammatory protein-2; CXCL-1, C-X-C motif chemokine ligand 1; ICAM-1, intracellular adhesion molecule-1; Arg-II, arginase-II; IL, interleukin; VCAM-1, vascular cell adhesion molecule-1; NF-κBIZ, NF-κB inhibitor zeta.

**Figure 2 ijms-24-16912-f002:**
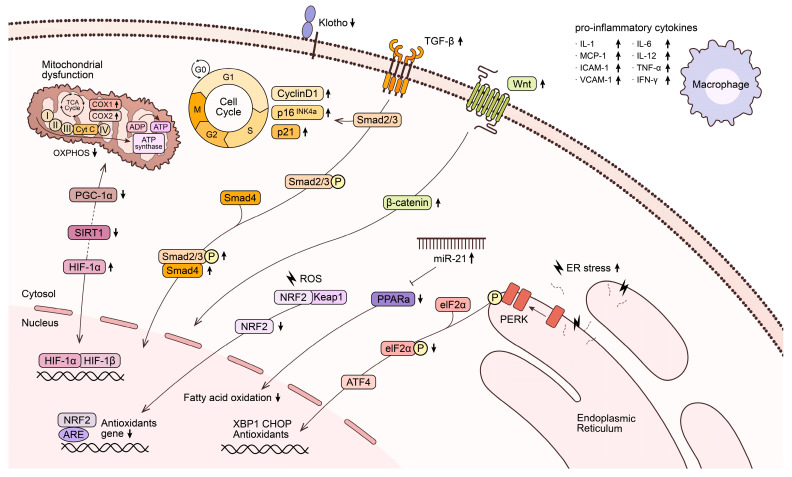
Molecular mechanisms in aged tubule. Abbreviations: COX-1, cyclooxygenase 1; COX-2, cyclooxygenase2; ADP, adenosine diphosphateta, adenosine triphosphate; TCA, tricarboxylic acid; Cyt C, cytochrome C, PGC1α, PPAR-α coactivador-1; OXPHOS, oxidative phosphorylation; SIRT1, sirtuin 1; HIF-1α, hypoxia-inducible factor 1 subunit α; HIF-1β, hypoxia-inducible factor 1 subunit β; PPARα, peroxisome proliferator activated receptor α; TGF-β, transforming growth factor-β; Smad2/3, SMAD family member 2/3; Smad4, SMAD family member 4; NRF2, nuclear factor erythroid-2-related factor 2; ARE, antioxidant responsive element; PERK, protein kinase RNA-like endoplasmic reticulum kinase; eIF2α, eukaryotic initiation factor 2α; ATF4, activating transcription factor 4; XBP1, X-binding protein 1, CHOP, CCAAT-enhancer-binding protein homologous protein; ER, endoplasmic reticulum; IL, interleukin; TNF- α, tumor necrosis factor-α; IFN-γ, interferon γ.

**Figure 3 ijms-24-16912-f003:**
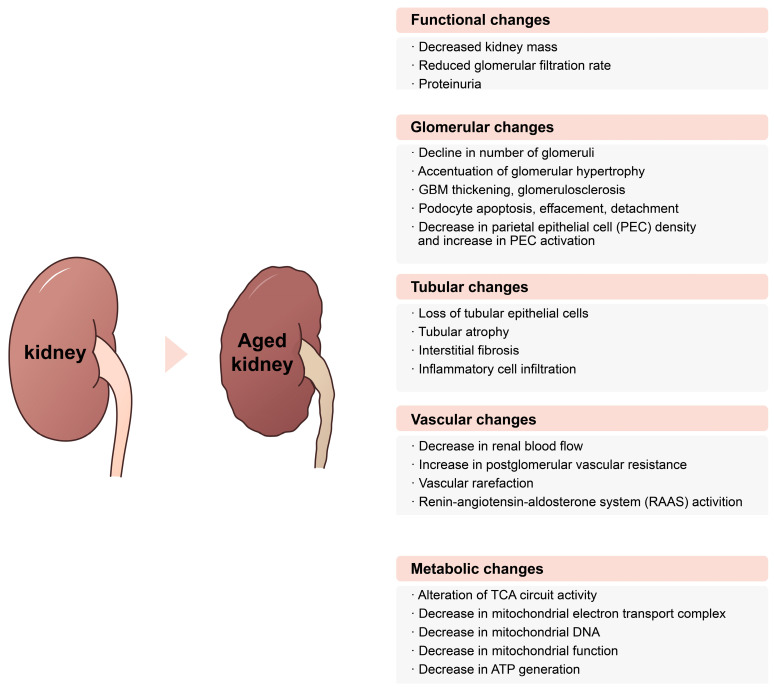
Functional, structural, and metabolic alterations in aging kidneys. Complex changes in various parts within the kidney are implicated in the decline of kidney function in aged kidneys. The changes occur sequentially with aging, and they are connected organically with the alterations happened in the adjacent tissues. Abbreviations: GBM, glomerular basement membrane; PEC, parietal epithelial cell; RAAS, renin-angiotensin-aldosterone system; TCA, tricarboxylic acid; DNA, deoxyribonucleic acid; ATP, adenosine triphosphate.

**Table 1 ijms-24-16912-t001:** CKD prevalence in the elderly population based on large-scale cohort studies.

No	Author (Study Year)	Region	Cohort	Elderly Population (≥65 Years)/Total Population	Age of Participants (Years)	CKD Definition	CKD Prevalence	Correlation with Aging			
1	Stevens et al. [14] (2010)	US	KEEP (2000–2008) NHANES (1999–2006) Medicare 5% sample (2006)	KEEP: 27,017/107,309 NHANES: 5538/41,474 Medicare 5% 2006: 1,236,946/1,479,818	≥65	KEEP, NHANES: eGFR < 60 mL/min/1.73 m^2^ (MDRD)oralbuminuria (ACR ≥ 30 mg/g) Medicare 5% sample: ICD-9-CM CKD	KEEP: 43.6% NHANES: 44.2% Medicare 5%: 6.5%	**age (y)**	**KEEP**	**NHANES**	**Medicare**
								65–74 (%)	61.9	55.0	50.8
								75–59 (%)	20.9	20.4	20.7
								≥ 80 (%)	17.1	24.6	28.5
2	Nicola et al. [15] (2015)	Italy	OEC/HES-CARHES	-/7552	35–79	Early stages CKD G1/G2 A2–3 Advanced stage CKD (≥G3): eGFR < 60 mL/min/1.73 m^2^ (CKD EPI)	G1 & G2: 4.16% G3a-G5: 2.89%	CKD prevalence across age strata: 35–49 years: 2.7% 50–59 years: 3.4% 60–69 years: 8.7% 70–79 years: 17.0%			
3	Ebert et al. [16] (2016)	Germany	BIS with AOK-Nordost sample	2069/2069 (≥70 years of age)	≥70	eGFR < 60 mL/min/1.73 m^2^ (CKD-EPI)	39.3%	eGFR(CKD EPI) < 60 in each age group: 70–74 years: 16.6% 75–79 years: 28.3% 80–84 years: 47.3% 85–89 years: 57.9% ≥90 years: 76.2%			
4	Bruck et al. [17] (2016)	Europe	19 general population-based studies from 13 European countries	64,137/189,171	20–74	eGFR < 60 mL/min/1.73 m^2^ (CKD EPI)	1.0–5.9%	CKD 3–5 prevalence in each age group: 20–44 years: 0.1–0.6% 45–64 years: 0.8–6.4% 65–74 years: 4.1–25.5%			
5	Arora et al. [18] (2017)	USA	US Department of Veterans Affairs, VISN 2	9212/180,533	≥20	eGFR 15–59 mL/min/1.73 m^2^	9.9%	**CKD stage**	**Predicted progression rate**		
									**<1%**	**1–4%**	**>4%**
								3a	49.4%	48.3%	2.3%
								3b	61.8%	37.7%	0.5%
								4	69.4%	30.6%	0
6	Polkinghorne et al. [19] (2019)	USA Australia	ASPREE	17,762/17,762	≥65	eGFR < 60 mL/min per 1.73 m^2^ (CKDEPI,BIS1) or UACR ≥ 3 mg/mmolwitheGFR ≥ 60 mL/min/1.73 m^2^	27%	CKD prevalence by age groups: 65–69 years: 3% 70–74 years: 44% 75–79 years: 28% 80–84 years: 19% ≥85 years: 7%			
7	Kibria, Crispen [20] (2020)	USA	NHANES (2003–2018)	-/39,569	≥20	eGFR < 60 mL/min/1.73 m^2^ (CKD EPI) or UACR ≥ 30 mg/g	18%	Age with CKD Prevalence (OR) 20–39 years: reference 40–59 years: 1.5 ≥60 years: 5.9			
8	Li et al. [21] (2023)	Japan	6th NDB Open Data Japan 2019 database	-/approximately 29.4 million	40–74	eGFR < 60 mL/min/1.73 m^2^,dipstickproteinuria ≥ 1+	CKD G3-G5: 11.1% Proteinuria ≥ 1+: 3.72%	Age was positively correlated with prevalence of lower eGFR (r = 0.716, *p* < 0.0001) Correlation coefficient between age and prevalence of proteinuria was very low (r = 0.196, *p* < 0.001)			

Abbreviations: CKD: Chronic Kidney Disease, KEEP: Kidney Early Evaluation Program, NHANES: National Health and Nutrition Examination Survey, eGFR: estimated Glomerular Filtration Rate, MDRD: Modification of Diet in Renal Disease, ACR: Albumin-Creatinine Ratio, ICD-9-CM: International Classification of Diseases, Ninth Revision, Clinical Modification, OEC/HES: Cardiovascular Epidemiology Observatory/Health Examination Survey, CARHES: Cardiovascular risk profile in Renal patients of the Italian Health Examination Survey, CKD-EPI: Chronic Kidney Disease Epidemiology Collaboration, BIS: Berlin Initiative Study. AOK-Nordost: Allgemeine Ortskrankenkass-Nordost, ASPREE: ASPirin in Reducing Events in the Elderly, VISN: Veterans Integrated Services Networks, UACR: urinary albumin-creatinine ratio, NDB: National DataBase.

## Data Availability

Not applicable.

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
