# Peer review of "Molecular Mechanisms Associated with Aging Kidneys and Future Perspectives"

_ijms, 2023, doi:10.3390/ijms242316912_

Round 1

Reviewer 1 Report

Comments and Suggestions for Authors

The authors discuss the functional and molecular mechanisms in aging kidneys. They separately describe the glomerular, tubular, vascular and metabolic changes in detailed. However, many knowledge are already known to most researcher. Moreover, some issues need to be addressed.

(1) The authors discussed a lot about the measurement of aging kidneys. However, these contents may not be within the scope of IJMS. So I recommend this part should be removed.

(2) More details should be added to provide complete information in Figure 1 as described in Section 3.2..

(3) Figure 3 provides a overview of aging kidneys from four different points. The figure should be provided in advance, rather than put at the end.

(4) The Future Perspectives section is too broad and only convey limited knowledge.

(5) The manuscript should have a conclusion Section.

(6) As mentioned in Section 5, some single cell and spatial omics data have been reported, like PMID: 37675124, PMID: 37827693, etc. How do these single cell studies contribute to our improvement in investigating aging kidney? 

Comments on the Quality of English Language

Minor editing of English language required

Reviewer 2 Report

Comments and Suggestions for Authors

In the paper entitled "Molecular Mechanisms associated with aging kidneys and Future Perspectives“, the authors provide a complex overview of the changes in kidney ageing in the human population and the molecular mechanisms that have been studied in animal models. The review is well written with up-to-date references. It represents a fruitful scientific work in the field of nephrology. There is not much to criticise. However, as the paper combines data from patients and molecular analyses of animal models, it would be helpful to mention this in the text as the reader may be confused.

Minor comment:

Page 1: 37-38 The units for weight should be “g“ and not “gm“. This should be corrected.

Round 2

Reviewer 1 Report

Comments and Suggestions for Authors

Thanks for the authors' efforts.

My questions have been well addressed, and the manuscript has been greatly improved.